# Role of Exosomes in Immunotherapy of Hepatocellular Carcinoma

**DOI:** 10.3390/cancers14164036

**Published:** 2022-08-21

**Authors:** Bao-Wen Tian, Cheng-Long Han, Zhao-Ru Dong, Si-Yu Tan, Dong-Xu Wang, Tao Li

**Affiliations:** 1Department of General Surgery, Qilu Hospital, Shandong University, Jinan 250000, China; 2Department of Hepatobiliary Surgery, The Second Hospital of Shandong University, Jinan 250000, China

**Keywords:** exosomes, hepatocellular carcinoma, immunotherapy, immune checkpoint inhibitor, tumor vaccine, adoptive cell therapy, efficacy, resistance mechanism, predictive effect

## Abstract

**Simple Summary:**

Hepatocellular carcinoma is one of the most lethal malignancies, having a significantly poor prognosis. Immunotherapy, as an emerging tumor treatment option, provides new hope for many cancer patients. However, a large proportion of patients do not benefit from immunotherapy. As a critical cell-to-cell communication mediator in the tumor immune microenvironment, exosomes may play a unique role in hepatocellular carcinoma immune response and thus affect the efficiency of immunotherapy. In this review, we discuss related research on the roles of exosomes in the current immunotherapy resistance mechanism of hepatocellular carcinoma. Furthermore, we also clarify the excellent predictive value of exosomes and the roles they play in improving immunotherapy efficacy for hepatocellular carcinoma patients. We hope that our review can help readers to gain a more comprehensive understanding of exosomes’ roles in hepatocellular carcinoma immunotherapy.

**Abstract:**

Hepatocellular carcinoma (HCC) is one of the most lethal malignancies, having a significantly poor prognosis and no sufficiently efficient treatments. Immunotherapy, especially immune checkpoint inhibitors (ICIs), has provided new therapeutic approaches for HCC patients. Nevertheless, most patients with HCC do not benefit from immunotherapy. Exosomes are biologically active lipid bilayer nano-sized vesicles ranging in size from 30 to 150 nm and can be secreted by almost any cell. In the HCC tumor microenvironment (TME), numerous cells are involved in tumor progression, and exosomes—derived from tumor cells and immune cells—exhibit unique composition profiles and act as intercellular communicators by transporting various substances. Showing the dual characteristics of tumor promotion and suppression, exosomes exert multiple functions in shaping tumor immune responses in the crosstalk between tumor cells and surrounding immune cells, mediating immunotherapy resistance by affecting the PD-1/PD-L1 axis or the anti-tumor function of immune cells in the TME. Targeting exosomes or the application of exosomes as therapies is involved in many aspects of HCC immunotherapies (e.g., ICIs, tumor vaccines, and adoptive cell therapy) and may substantially enhance their efficacy. In this review, we discuss the impact of exosomes on the HCC TME and comprehensively summarize the role of exosomes in immunotherapy resistance and therapeutic application. We also discuss the potential of exosomes as biomarkers for predicting the efficacy of immunotherapy to help clinicians in identifying HCC patients who are amenable to immunotherapies.

## 1. Introduction

Primary liver cancer is the sixth-most commonly occurring cancer and third leading cause of cancer death worldwide, of which hepatocellular carcinoma (HCC) accounts for 75–85% of the total liver cancer burden [1]. Due to the lack of distinguishing characteristics and effective screening methods in the early stage, the majority of HCC cases are diagnosed at the advanced stage, which leads to the overall survival of most HCC patients being less than 5 years [2,3]. Despite numerous measures in the field of HCC treatment, effective treatment options for advanced HCC are still limited [4]. With the deepening understanding of the tumor microenvironment (TME) in HCC, immunotherapy—especially immune checkpoint inhibitors (ICIs)—has brought new hope for improving the prognosis of advanced HCC. ICIs mainly include monoclonal antibodies against cytotoxic-T-lymphocyte-associated protein 4 (CTLA-4), programmed cell death protein 1 (PD-1), and its ligand-programmed cell death ligand 1 (PD-L1), which block these immune checkpoints, thereby promoting the T cell-mediated antitumoral immune responses [5,6,7,8]. Other immunotherapies, including cancer vaccines and adoptive cell therapy, have also shown great potential for clinical application to treat advanced HCC [9,10]. Nevertheless, numerous HCC patients do not benefit from immunotherapy, which has seriously restricted its application [11,12].

At present, the relationship between immunotherapy and exosomes is a new hotspot in HCC research that may improve our understanding of the resistance mechanism of immunotherapy. Exosomes are small, single-membrane secreted organelles ranging in diameter from 30 to 150 nm [13]. They are formed by the invagination of the cell membrane of early sorting endosomes, which eventually grow into multivesicular bodies [14,15]. As a type of extracellular vesicle (EV), exosomes contain a variety of nucleic acids, proteins, lipids, and metabolites which can reflect, at least in part, those of their parental cells [16]. By transporting these components, exosomes participate in numerous biological processes, including antigen presentation, apoptosis, inflammation, and intercellular signaling, etc. [17]. Exosomes, particularly the proteins and noncoding RNA in them, play a crucial role in tumor growth, metastasis, angiogenesis, and invasion [18] and are closely linked with tumor immunotherapy [19]. Recent studies have demonstrated that tumor-cell-derived exosomes (TEXs) are also involved in TME remodeling, tumor progression, and drug resistance, which brings challenges for the effective treatment of HCC [20,21]. These complex biological roles enable exosomes to be applied for early diagnosis, treatment, and prognostic prediction of HCC [22,23,24], and research on exosomes is expected to help clinicians to identify suitable HCC patients for immunotherapy and improve the efficacy of immunotherapy.

In this review, we searched the literature with high quality and strong relevance to our topic to discuss the impact of exosomes on the HCC TME and comprehensively summarize the role of exosomes in drug resistance, therapeutic application, prognostic biomarkers, and efficacy prediction of immunotherapy for HCC.

## 2. Influence of Exosomes on Tumor Microenvironment of HCC

The HCC TME is an elaborate immunosuppressive microenvironment composited of different types of cells, growth factors, proteolytic enzymes, extracellular matrix proteins, and cytokines [25]. In HCC TME, not only immune cells but also non-immune cells are involved in tumorigenic processes, including tumor proliferation, invasion, and metastasis [26,27]. Recent studies have highlighted the critical role of exosomes for cell-to-cell communication in the HCC TME, which are recognized as a key factor in tumor progress and may be closely associated with resistance to immunotherapy in HCC patients [28,29].

### 2.1. Exosomes on Cell-to-Cell Communication in HCC TME

The influence of exosomes on cell-to-cell communication in HCC TME is summarized in Figure 1 and Appendix A.

Tumor-cell-derived exosomes can be directly transported into CD8+ T cells, tumor-infiltrating T cells, or regulatory T cells to inhibit their anti-tumor function or modulate their cellular behaviors [30,31,32,33]. Furthermore, HCC-derived exosomes can induce nature killer (NK) cell activation, attenuate NK cell cytotoxicity, and arrest them to secret cytokines [34,35,36]. HCC-derived exosomes can effectively activate dendritic cells (DCs) due to their ability to carry HCC antigens [37], while DC-derived exosomes (DEXs) can potentiate anti-tumor immune responses against HCC by contributing to the activation of CD8+ T cells and reshaping the TME [38,39,40]. Depending on the substances they carry, HCC-derived exosomes show apparently contradictory roles on macrophages. They can directly influence the expression of macrophage cell surface receptors or membrane-associated signaling molecules [32,41,42,43,44] and can affect the differentiation of macrophages into two different directions: M1 (anti-tumorigenic) [45] or M2 (pro-tumorigenic) macrophages [46,47,48,49]. Furthermore, macrophage-derived exosomes can also affect HCC development or other immune cell functions [50,51,52,53,54,55,56,57]. High-mobility group box1 (HMGB1) is an evolutionarily conserved DNA-binding nuclear protein which has been demonstrated to be expressed on tumor-derived exosomal membranes. Recent study has revealed that HCC-derived exosomal HGMB1 can foster HCC immune evasion by promoting TIM-1+ regulatory B cell expansion [58]. Exosomes derived from myeloid-derived suppressor cells, a type of special cell in tumor immunity, play important immunoregulatory roles in cancers, and tumor-derived exosomes also regulate the function of myeloid-derived suppressor cells [59,60,61]. Unfortunately, no relevant studies have been undertaken in HCC.

There have also been numerous studies on the relationship between exosomes and non-immune cells in the HCC TME. HCC is a typical hypervascular tumor, HCC-derived exosomes can promote angiogenesis by targeting vascular endothelial cells [62,63], and exosomal CXCR4 has also been proved to promote HCC lymphangiogenesis [64]. Cancer-associated fibroblast-derived exosomes can affect the development, invasion, drug-resistance, and metabolism of HCC [65,66,67], and adipocyte-derived exosomes also have similar tumor-promoting effects [68,69]. Furthermore, HCC-derived exosomes can be transported to normal hepatocytes or other tumor cells to enhance motile ability [70,71], stimulate epithelial–mesenchymal transition of peripheral cells, and further promote HCC invasion [72,73]. In contrast, mesenchymal-stem-cell-derived exosomes can inhibit the malignant behavior of HCC to some degree by transporting some microRNAs, making them potentially useful for HCC treatment [74,75].

### 2.2. Exosomes on Immunotherapy Resistance of HCC

There are several vital exosome-associated mechanisms that may partly explain why many HCC patients show resistance to ICIs (Figure 2). ICIs, mainly against CTLA-4, PD-1, and PD-L1, not only invigorate T cells but also activate other cells involved in innate and adaptive immune response, all of which function together to inhibit tumors [76,77]. CD8+ T cells have great potential in ICI therapy and adoptive T cell therapy, but they are often dysfunctional in tumors [78,79]. In the HCC TME, exosomes can affect the function of CD8+ T cells through multiple pathways, which may lead to immunotherapy resistance. PD-1 is a vital co-inhibitory receptor which is primarily expressed on the surface of antigen-stimulated T cells. However, PD-L1 is upregulated on tumor cells and antigen-presenting cells in the TME and is bound to PD-1 on activated T cells and dampens anti-tumor immunity by counteracting T-cell-activating signals [80,81,82]. Exosomes may increase the expression of PD-1 or PD-L1, which augments the effective dose of ICIs. TEXs can directly impact the PD-1/PD-L1 expression of immune cells in the TME by carrying PD-L1 or indirectly impact this by regulating the PD-1/PD-L1 axis in adjacent immune cells, thereby affecting the efficacy of ICIs [83,84]. Spleen deficiency (SD) can also upregulate the expression of exosomal CTLA-4 and PD-1 [85].

ICIs can block PD-L1, but exosomes can convey cargos stably, which may help exosomal PD-L1 to avoid being blocked by ICIs [86]. Exosomal PD-L1 has the same membrane topology as PD-L1 on the surface of immune cell, which can directly suppress the function of T cells by binding to PD-1, thereby inducing HCC patients’ resistance to ICIs [83]. Exosomes containing PD-L1 are a major means by which HCC cells can have a dramatic influence on T cell function. Recently, Chen et al. demonstrated that Golgi membrane protein 1 in HCC can facilitate PD-L1 trafficking to exosomes by suppressing Rab27b in the trans-Golgi network (TGN) area [41]. Meanwhile, norcholic acid can elevate the expression of exosomal PD-L1 through inhibiting farnesoid X receptor signaling [87]. Furthermore, the HGMB1-driven RNA–RNA crosstalk network facilitated HCC cell glutamine metabolism, which could compromise the efficacy of immunotherapy through mTORC1-P70S6K-dependent PD-L1 production and PD-L1+ exosome activity [88]. These PD-L1-carrying exosomes can be taken up by macrophages and upregulate their PD-L1 expression, further inhibiting the anti-tumor function of CD8+ T cells [41]. Additionally, it is worth noting that in a study of melanoma treatment with pembrolizumab, an increase in the level of exosomal PD-L1 during the treatment may reinvigorate the IFN-γ to produce CD8+ T cells. However, the high pre-treatment levels of exosomal PD-L1 were associated with T cell exhaustion, meaning that it could not be reinvigorated by PD-1 inhibitors [89].

This is also a common way for HCC-derived exosomes to indirectly influence T cell function through surrounding cells. Endoplasmic reticulum stress contributes HCC cells releasing miR-23a-3p-enriched exosomes, which inhibit PTEN and in turn activate the AKT pathway, resulting in high PD-L1 expression in macrophages [42]. HCC-derived LOXL4, which is secreted through exosomes and primarily localized within macrophages, can strengthen PD-L1 expression in macrophages relying on interferon-mediated signal transducer and activator of transcription-dependent PD-L1 activation [43], thereby dampening the function of CD8+ T cells. Furthermore, HCC-derived exosomal circTMET181 sponges miR-488-3p and upregulates CD39 expression in macrophages. Macrophage-derived CD39 and HCC cell-derived CD73 synergistically activate the ATP–adenosine pathway, which in turn degrades extracellular ATP into adenosine. This process generates a large volume of adenosine, which impairs CD8+ T cell function and leads to anti-PD1 resistance [44]. In addition, the up-regulation of 14-3-3ζ protein can restrain the anti-tumor immunity of tumor-infiltrating T cells in TME, which may be due to CD8+ T cell exhaustion [30], and HCC cells can also release exosomal PCED1B-AS1 to inhibit recipient T cells [32].

Exosomes inhibit the anti-tumor function of T cells through the various above-mentioned mechanisms, thereby affecting the efficacy of ICIs. In addition, B cells are central to humoral immunity. HCC-derived exosomes promote TIM-1+ regulatory B cell expansion through exosomal HMGB1-TLR2/4-MAPK pathways and further lead to the overproduction of the immunosuppressive cytokine IL-10, which results in a marked inhibition of CD8+ T cell activity [58]. The effect of exosomes on NK cells, macrophages and other immune cells may also cause immunotherapy resistance, but these mechanisms require further in-depth research. In addition, the research on exosomes in other immunotherapies for HCC (such as tumor vaccines or adoptive cell therapies) is also limited. More related research will help us to better understand the role of exosomes in immunotherapy resistance.

## 3. Exosomes and Efficacy of HCC Immunotherapy

Exosomes are biologically active lipid bilayer nano-sized vesicles that not only can deposit cargos stably but also are convenient for storage and transportation [86]. Their structural characteristics give them great potential to potentiate the efficacy of anti-tumor immunotherapies [9], such as ICIs, tumor vaccines, and adoptive cell therapy. At the same time, due to the crucial role of exosomes in immunotherapy resistance, drugs developed for exosomes are also expected to potentiate the efficacy of HCC immunotherapy (Table 1).

### 3.1. Role of Exosomes in Improving ICI Efficacy

The role of exosomes in improving ICI efficacy is summarized in Figure 3. To improve the efficacy of ICIs in HCC, it may be of paramount importance to control the expression of exosomal PD-L1 [41,90]. It is suggested that interference with HMGB1 and RICTOR can inhibit the production of PD-L1-contating exosomes, which may be achieved by administering miR-200a/200b/429 mimics [88]. However, it should be noted that HMGB1 appears to play paradoxical roles during the development and therapy of cancer, and its value as potential target for therapies is worthy of further study. The effect of eliminating exosomal PD-L1 on ICI treatment of HCC remains to be further validated. In addition, exosomal PD-L1 can be eliminated not only by suppressing the generation and secretion of tumor cells but also by selectively removing the circulating exosomes from the bloodstream using hemofiltration [91]. Although hemofiltration can selectively remove immunosuppressive exosomes without removing other exosomes responsible for the normal intercellular communication, some preclinical studies have not found obvious immune-related adverse events. However, it should be noted that there may be the risk of serious immune-related adverse events after eliminating circulating exosomal PD-L1 since it plays a role in mediating systemic immunosuppression.

Inhibiting the expression of PD-1/PD-L1 on the cell surface of immune cells by exosomes also has significant therapeutic potential. The efficacy of ICIs may be enhanced by downregulating the expression of PD-L1 on macrophages, which may be accomplished by the administration of exosomes derived from HCC cells treated with melatonin [92]. Tumor-derived exosomal miR-15a-5p may also enhance the efficacy of ICIs by inhibiting PD-1 expression in CD8+ T cells. Furthermore, the therapeutic response to PD-1 inhibitor can be augmented by targeting β-catenin using exosome-encapsulated small interfering RNA [93]. Finally, Teng et al. clarified that the simultaneous use of DC vaccine can also augment the efficacy of PD-1 inhibitor in HCC mice [94]. Notably, DCs used in this study were DCs pulsed with HCC cell lysates, which are less effective than TEX-pulsed DCs (DC-TEX) [37,95]. Thus, a combination treatment using both DC-TEX and PD-1 inhibitor may be a more ideal choice for HCC patients. Another study, to some extent, supports the above view that a combined use of DC-TEX and PD-1 inhibitor could enhance the efficacy of sorafenib, but treatment with either DC-TEX or PD-1 inhibitor alone did not [96]. As we illustrate before, SD may restrict the efficacy of ICIs by upregulating the expression of exosomal CTLA-4 and PD-1 [85]. Thus, for HCC patients with SD, the efficacy of ICIs may be reinforced by treating SD through the administration of tonifying traditional Chinese medicine. There was a novel investigation into modifying the exosomes to display monoclonal antibodies on the exosome surface to treat epithelial malignancies [97], indicating that we may strengthen the efficacy of ICIs for HCC patients by reprogramming exosomes to display monoclonal antibodies against PD-1, PD-L1, or CTLA-4 on the exosome surface.

### 3.2. Role of Exosomes in Improving Tumor Vaccine Efficacy (Figure 4A)

According to our previous study, we noted a marked disparity in the efficacy of different tumor vaccines: DC vaccines hold a significantly greater efficacy than other types of tumor vaccines for HCC at present, requiring further attention [98]. In light of the latest research, there are several strategies for reinforcing the efficacy of DC vaccines. DC-TEX possesses a better efficacy than DC lysates, and DCs pulsed with TEXs that are coated with the functional domain of HMGN1 may possess the best efficacy [99]. DEXs can enhance the antitumor effect of immune cells in the HCC TME. High-purity DEXs can promote the proliferation of naïve T cells and differentiate to cytotoxic T lymphocytes to exert anti-tumor effects against HCC [39]. Similarly, exosomes derived from AFP-expressing DCs (DEX-AFP) can significantly increase the number of CD8+ T cells, elevate levels of γ-interferon and interleukin-2, and reduce the number of CD25+Foxp3+ regulatory T cells [38]. Given these outcomes, DEXs, particularly DEX-AFP, may be promising for fully or partly substituting mature DCs to serve as a cell-free tumor vaccine. It is also noteworthy that when combined with microwave ablation, there was no statistically significant difference in the anti-tumor effects between DC and DEX [100].

As another kind of tumor vaccine with great potential, the rationale of virus vaccines is that an oncolytic virus can target tumor cells with high specificity, lyse tumor cells, and elicit the host anti-tumor immunologic response [9]. As one of the most commonly applied oncolytic virus, type V adenovirus (Ad5) still has many limitations in its clinical application. Zhang et al. used extracellular vesicle mimetic technology to design EVM/VSV-G Ad5-P which can avoid being neutralized by anti-Ad5 antibodies [101]. This might provide a new strategy based on exosomes to improve the effectiveness of virus vaccines.

### 3.3. Role of Exosomes in Improving Adoptive Cell Therapy (Figure 4B)

As a direct immunotherapy, adoptive cell therapy predominantly incorporates chimeric antigen receptor T (CAR-T) cells, cytokine-induced killer cells, NK cells, and tumor-infiltrating lymphocytes [9]. CAR-T cell therapy is the most rapidly developing therapeutic strategy for adoptive cell therapy so far and has achieved remarkable accomplishments in the field of hematological malignancies [102], and GPC3-targeted CAR T cell therapy may be applicable to HCC patients [103]. The simultaneous use of anti-PD-L1 antibodies may potentiate the efficacy of CAR-T cell therapy [102]. The efficacy of CAR-T cells therapy may also be augmented by co-deploying peptide antigens with RN7SL1-containing exosomes [104]. However, there remains a troublesome issue, in that CAR-T cells are difficult to store and transport [102], which seriously restricts the administration of CAR-T cell therapy. CAR-containing exosomes derived from CAR-T cells are not only safe but also hold potential anti-tumor effects [105], are expected to solve the above problems, and deserve further attention. The biogenesis and secretion of CAR-T-cell-derived exosomes may be positively modulated by some T lymphocyte activation enhancers (i.e., DGK inhibitors, such as R59949), which may constitute a promising strategy for CAR-T-cell-derived exosome-based immunotherapies [106].

In HCC patients, the development of HCC is usually in parallel with the exhaustion of immune cells, such as diminished numbers of NK cells [107] and the functional impairment of T cells [108], which may be fully or partly associated with the recurrence of HCC, particularly advanced-stage HCC after treatment. Therefore, potentiating the efficacy of immunotherapy for HCC patients may also be achieved by the administration of efficient immune cells, such as NK cells, M1 cells, and so on. Efficient NK cells may be elicited by administrating HSP-bearing exosomes [109], and the cytotoxicity of NK cells may be reinforced by the administration of exosomes derived from hepatoma G2 cells modified by the epigenetic drug MS-275 [110]. M1 macrophages polarization may be heightened by the administration of exosomes synergized with PIONs@E6 [111], exosomes derived from M1 macrophages transfected with NF-κB p50 siRNA and miR-511–3p, and surface-modified with IL4RPep-1 [112]. Moreover, engineered exosomes delivering an antisense oligonucleotide targeting STAT6 can selectively silence STAT6 expression in tumor-associated macrophages (TAMs), which may help us to reprogram TAMs toward a pro-inflammatory M1 phenotype to heighten CD8+ T cells efficacy in the TME [113]. In addition to these types of cells, after a comprehensive understanding of exosomes, the administration of mast cells stimulated by hepatitis C virus E2 envelope glycoprotein or exosomal shuttle microRNAs from those cells can inhibit the metastasis of HCC cells, which may ultimately lead to the longer survival of HCC patients [114].

### 3.4. Role of Exosomes in Other Therapies (Figure 4C)

Exosomes can also enhance the efficacy of HCC immunotherapy through other pathways. β-catenin mutation is well known as one of the most frequent mutations in HCC, and the activation of Wnt/β-catenin signaling is closely associated with HCC immune evasion. In addition, EVs containing small interfering RNA targeting oncogenic β-catenin can suppress the growth of tumors. Given these outcomes, the use of both PD-1 inhibitor and β-catenin siRNA can exert a synergistic effect to promote CD8+ T cells’ infiltration into the HCC TME [93]. Another in vitro experiment concluded that the delivery of miR-125b-loaded EVs can evidently suppress HCC growth through the regulation of a series of targets in the p53 signaling pathway [115]. However, EVs also suffer from numerous issues with respect to biological security and target sensitivity. Zhang et al. considered that EVs from red blood cells can naturally accumulate in the liver and may diminish the systemic toxicity of delivered drugs. The mechanism behind this phenomenon is the liver environment, which induces macrophages to phagocytize these EVs in a C1q-dependent manner [116].

Mesenchymal stem cells (MSCs) are also considered to be a promising therapeutic approach for HCC treatment. As early as ten years ago, Ma and colleagues elucidated that the homologous TEXs could enhance the migratory capacity of bone MSCs (BMSCs) and promote their anti-tumor effects against HCC [117]. BMSC-derived exosomal miR-338-3p had also been proved to effectively inhibit HCC development by down-regulating EST1 [118]. In addition, adipose-derived MSCs exosomes might promote the anti-tumor response of NKT cells, and human umbilical cord MSCs exosomal miR-451a targeting ADAM10 could restrict the epithelial–mesenchymal transition of HCC cells [119,120]. These findings provide a solid foundation for the application of MSCs in clinical immunotherapy of HCC.

## 4. Role of Exosomes in Predicting the Efficacy of HCC Immunotherapy

### 4.1. Detection Methods and Technologies for Exosomes

Exosomes can be extracted from various body fluids, such as the blood, sputum, and urine, which are accessible by means of minimally invasive liquid biopsies [121,122]. Exosomes were usually separated by ultracentrifugation [123], and there are well-designed commercial kits for isolating exosomes from urine or blood [124]. The expression of exosomal miRNAs can be evaluated by RT-PCR, and the expression of exosomal proteins can be assessed by mass spectrometry [123] or enzyme-linked immunosorbent assay (ELISA). Acoustic tweezer techniques in combination with microfluidics can also be applied to obtain exosomes from whole blood samples [125] or saliva samples [126]. The advancement of these liquid biopsy techniques holds great promise to revolutionize the diagnosis, prognosis judgement and efficacy prediction of HCC (Table 2).

Although detection technologies of exosomes have been well-established, a critical clinical-related issue naturally arises regarding which exosomes (from which type of body fluid) are the best exosomes for predicting the efficacy of HCC immunotherapy. Following YUSUF’s study, although both serum-derived and ascites-fluid-derived exosomal miRNA may be regarded as circulating biomarkers for HCC patients, the exact predictive values of them may be different [127], which indicates that exosomes from different body fluids may hold varying predictive value in HCC patients. Nakashiki et al., pointed out that blood exosomes may not be suitable for reflecting the liver state because they contain other systemically circulating exosomes. In contrast, bile exosomes, which are secreted directly by the liver, may harbor the best prognostic significance for liver disease [128]. From this outlook, bile exosomes may be the most suitable exosomes for predicting the efficacy of HCC immunotherapy. Further prospective high-quality investigations are warranted.

### 4.2. The Predictive Value of Exosomal PD-L1 for HCC Immunotherapy

Tumor immunotherapy has changed the therapeutic landscape for HCC patients. However, most patients do not benefit from immunotherapy. Therefore, identifying predictive biomarkers of the clinical response to ICIs is crucial for screening appropriate candidates for immunotherapy. Currently, numerous studies have focused on whether exosomal PD-L1 can be used as a biomarker to predict the efficacy of immunotherapy, and its predictive value was initially validated in melanoma [129]. Likewise, exosomal PD-L1 can predict worse prognosis in gastric cancer patients and reflect their immune states [130]. Moreover, a previous study which enrolled 44 patients with several types of advanced cancers hinted that exosomal PD-L1 can serve as the predictive biomarker for efficacy of anti-PD-1 therapy [131]. Though soluble PD-L1 has been proven to be a good predictor of the recurrence and survival of HCC [132], recent studies found that exosomal PD-L1, but not other forms of extracellular PD-L1, could be used to predict the prognosis or response to immunotherapy in various cancer types [133].

Until now, no studies have directly compared the predictive efficiency of exosomal PD-L1 with other clinical tumor markers, and some studies have found that exosomal PD-L1 cooperating with other tumor markers or immune modulators (such as CD28), may be a better predictive biomarker than exosomal PD-L1 alone [134]. Though PD-L1 expression on circulating tumor cells (CTCs) may be predictive of response to immunotherapy. However, the detection of CTCs remains a challenge because CTCs are usually very rare and different methods might enrich different CTC populations, which will affect the PD-L1 assessment. In contrast, exosomes are stable and abundant in plasma and can be quantified by ELISA. Therefore, exosomal PD-L1 may be the most promising potential biomarker in this area based on current relevant pan-cancer studies [133,135], and relevant studies are necessary to explore its application potential in HCC. In addition, there is also a homogeneous, low-volume, efficient, and sensitive quantitation method for exosomal PD-L1 which has unprecedented potential for immunotherapy response prediction [136]. More importantly, the pre-treatment and during-treatment level of exosomal PD-L1 may reflect distinct states of anti-tumor immunity. The high pre-treatment level of exosomal PD-L1 may reflect the fact that T cells will not be reinvigorated by anti-PD-1 therapy, and the high on-treatment level of exosomal PD-L1 may reflect that anti-PD-1 therapy successfully elicits anti-tumor immunity, which must be validated in HCC patients [89].

### 4.3. The Predictive Value of Other Exosomes for Immunotherapy

In addition to exosomal PD-L1, the exosomal genes *MYL6B* and *THOC2* may also be potential biomarkers for predicting the prognosis of immunotherapy, as they may influence the expression of immune checkpoint genes, such as *B7H5*, *CTLA4*, *PD1*, *B7H3*, and *TIM3* [137]. Zhu et al., also revealed an optimal gene set consisting of them for predicting HCC prognosis. Furthermore, another significant potential biomarker for predicting the efficacy of immunotherapy may be exosomal miR-143-3p, which can upregulate the expression of MARCKS in TAMs. MARCKS expression was considered significantly correlated with outcomes in HCC by influencing the polarization of M2 macrophages [138] that had tumor-promoting effects. The prediction of HGMB1 for HCC prognosis has been clinically confirmed [139]. HGMB1 in exosomes can affect the immunotherapy of HCC, so the detection of HGMB1 in exosomes is expected to predict the efficacy of HCC immunotherapy. More candidate predictive biomarkers for the efficacy of immunotherapy in HCC patients may be identified after a further understanding of the relationship between exosomes and immunotherapy in HCC patients.

## 5. Conclusions and Future Perspectives

More recently, the breakthroughs in HCC immunotherapies have shed new light on curative treatments for HCC patients. Nevertheless, the HCC immunotherapies are facing plenty of laborious challenges, and in-depth mechanistic exploration of exosome-mediated cell-to-cell communication in the HCC TME may assist us in overcoming them. Exosomes arise from almost any cell in the HCC TME, and their diverse compositions build an elaborate network of intercellular communication among distinct cells, and thereby can either promote HCC progress or exert anti-tumor effects. A deeper understanding of the roles of exosomes for HCC immunotherapies may contribute to constructing novel immunotherapeutic strategies and identifying HCC patients amenable to immunotherapies.

The mechanism of exosomes with respect to immunotherapy resistance substantially concentrates on two aspects. On the one hand, some of exosomes can alter the expression of PD-1/PD-L1 on the surface of immune cells by either direct or indirect means, ultimately leading to CD8+ effector cell depletion. On the other hand, some exosomes can be constraints on the anti-tumor function of adjacent immune cells in TME via other mechanisms, such as interfering with cell differentiation or driving cell exhaustion, which finally influence the T cells’ function.

Research on fortifying the efficacy of HCC immunotherapies with exosomes is also fully underway. We can enhance the efficacy of ICIs by enabling exosomes to carry drugs to downregulate the PD-L1 expression on the immune cells’ surface or by directly inhibiting the production of PD-L1-contating exosomes. It is generally thought that PD-L1 functions within the tumor bed, where cell-surface PD-L1 directly interacts with PD-1 on the surface of TILs. However, recent studies discover that tumor cells can secrete a vast majority of their PD-L1 on exosomes rather than present PD-L1 on their cell surface [90]. Exosomal PD-L1 enables cancer cells to evade anti-tumor immunity and appears to be resistant to anti-PD-L1 antibody blockade. Removal of exosomal PD-L1 inhibits tumor growth, even in models resistant to anti-PD-L1 antibodies. However, the degree to which exosomal versus surface PD-L1 in driving immunosuppression will vary between patients and cancer types and will be critical in deciding who is more likely to respond to therapy [90]. Therefore, measuring both cell surface and exosome presentation of PD-L1 should be considered in any therapeutic strategy, and it will be interesting to determine whether localized anti-exosomal therapy combined with systemic anti-PD-L1 blockade could synergize to induce a systemic immune response against tumor [90].

Exosomes also provide new ideas for the application of tumor vaccines, especially DC vaccines. DC-TEX may possess an augmented efficacy while DEXs may serve as cell-free vaccines with a similar efficacy, compared to traditional DC vaccines. Adoptive cell therapy is a novel treatment modality for HCC. Therapeutic exosomes can not only synergize with CAR-T therapy, but also affect the function of NK cells and M1 cells in HCC to relieve the immunosuppressive TME state. In addition to the abovementioned actions, the suppressive effects of MSC-derived exosomes on HCC are also expected to become a new direction for further immunotherapy. The exosomes developed according to their different properties for delivering immunotherapy drugs have important implications for precise drug localization, improving drug efficacy, and mitigating drug toxicity/adverse events. Despite there being much to be done before therapeutic exosomes enter the clinic, they are expected to become a crucial component of HCC immunotherapies.

As studies on the utilization of exosomes to anticipate the efficacy of HCC immunotherapies are relatively rare, exosomal PD-L1 may be a significant potential biomarker for the efficacy estimates of ICIs. Other exosome-containing substances, such as HMGB1, which exerts an essential role in mediating the HCC resistance to immunotherapies, also have great potential to some extent. The validation of the predictive value of these possible biomarkers in exosomes will require detailed high-quality prospective investigations. Despite the advancement in detection technology for exosomes, which body fluid’s exosomes can be used for prediction is still a critical issue waiting to be solved. Plasma/serum exosomes are more commonly used, while bile exosomes may be more specific. Therefore, finding suitable exosomal biomarkers for evaluating HCC immunotherapies will help clinicians in identifying HCC patients who are amenable to immunotherapies.

## Figures and Tables

**Figure 1 cancers-14-04036-f001:**
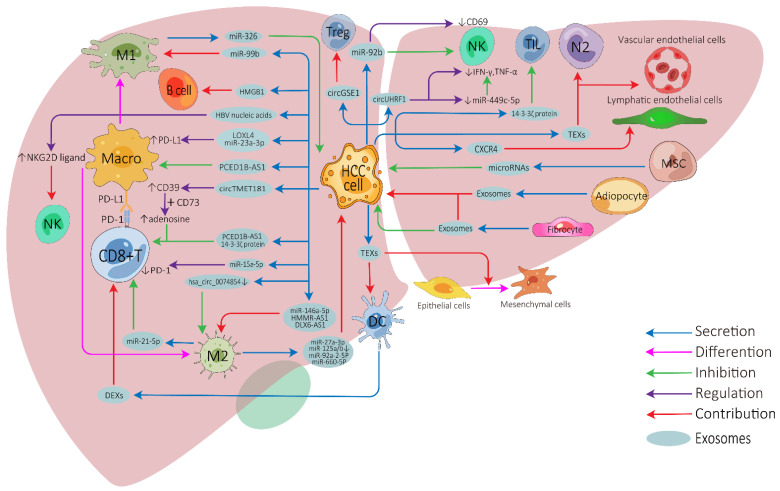
Exosome-mediated cell-to-cell communications between different cells in the HCC tumor microenvironment. HCC, hepatocellular carcinoma; Treg, regulatory T cell; NK, natural killer cell; Macro, macrophage; M1, M1-polarized macrophage; M2, M2-polarized macrophage; DC, dendritic cell; MSC, mesenchymal stem cell; N2, N2-neutrophil; TIL, tumor-infiltrating lymphocyte; TEXs, tumor-cell-derived exosomes; DEXs, DC-derived exosomes; PD-1, programmed cell death protein 1; PD-L1, programmed cell death ligand 1.

**Figure 2 cancers-14-04036-f002:**
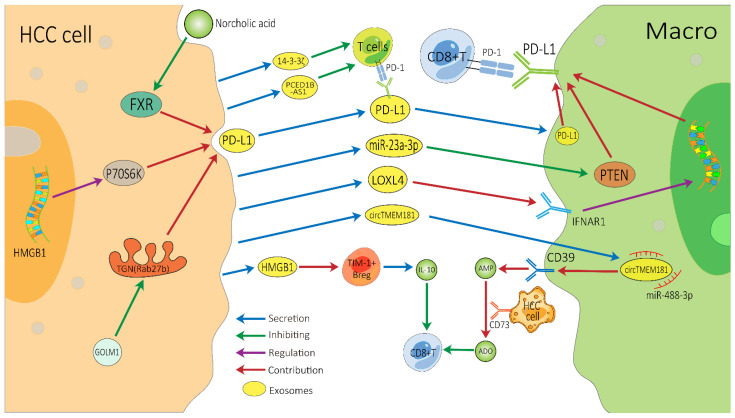
Possible exosome-associated resistance mechanisms for immunotherapy of HCC. TEXs can directly dampen the function of CD8+ T cells by carrying PD-L1 or indirectly dampen it by impacting other adjacent immune cells, thereby inducing HCC’s resistance to immunotherapy. HCC, hepatocellular carcinoma; Macro, macrophage; PD-1, programmed cell death protein 1; PD-L1, programmed cell death ligand 1; TEXs, tumor-cell-derived exosomes; FXR, farnesoid X receptor; GOLM1, Golgi membrane protein 1; ADO, adenosine; IL-10, interleukin-10.

**Figure 3 cancers-14-04036-f003:**
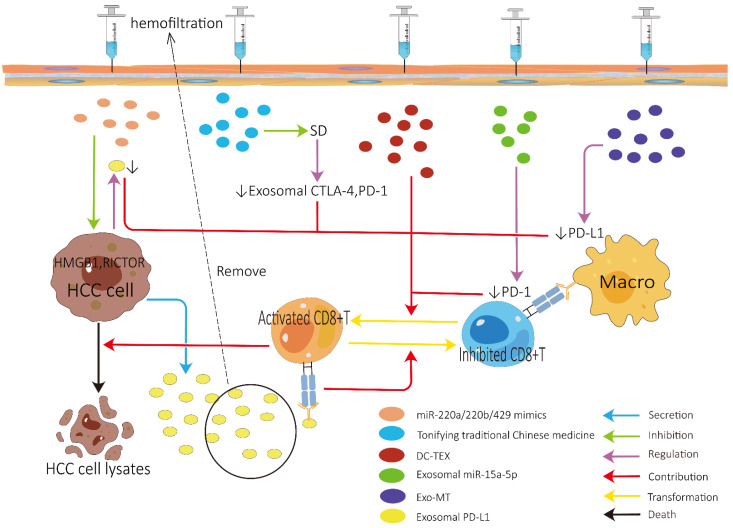
Potential role of exosomes in improving the efficacy of immune checkpoint inhibitors. HCC, hepatocellular carcinoma; Macro, macrophage; PD-L1, programmed cell death ligand 1; CTLA-4, cytotoxic-T-lymphocyte-associated protein 4; PD-1, programmed cell death protein 1; DC-TEX, tumor-cell-derived exosomes-pulsed DCs; SD, spleen deficiency; Exo-MT: Exosomes derived from HCC cells treated with 0.1 mM melatonin.

**Figure 4 cancers-14-04036-f004:**
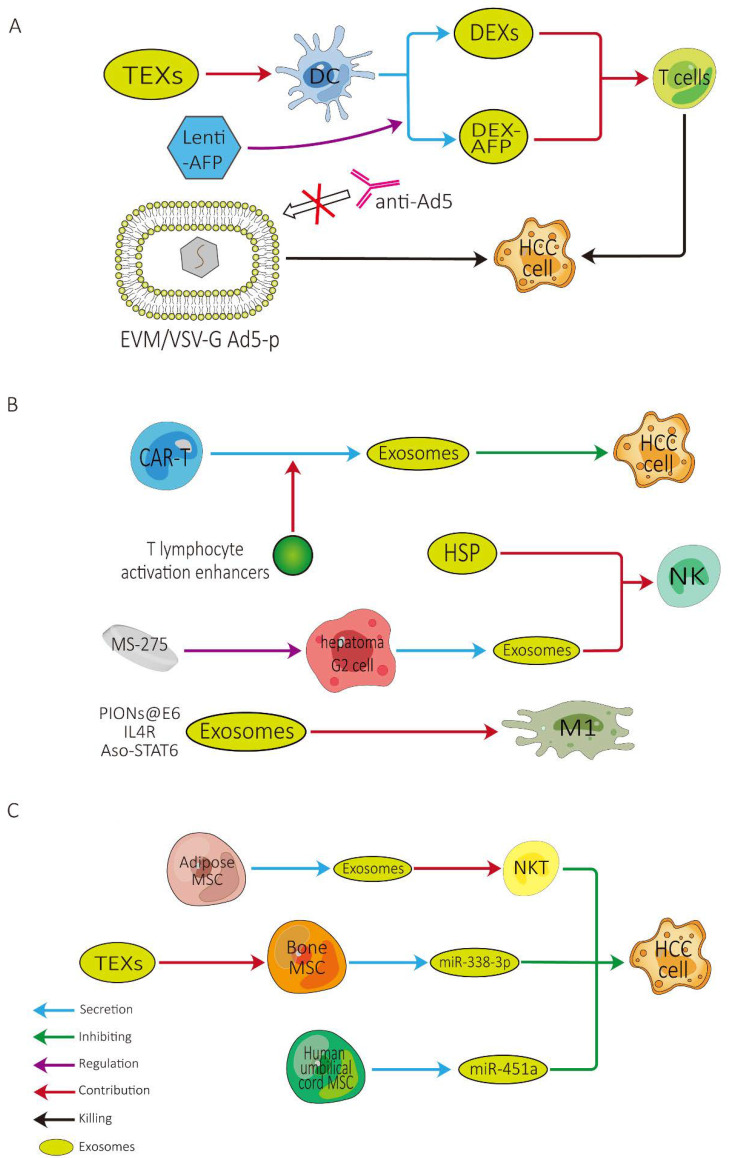
Potential role of exosomes in improving the efficacy of (**A**) tumor vaccine therapy; (**B**) adoptive cell therapy; and (**C**) mesenchymal stem cell therapy. HCC, hepatocellular carcinoma; TEXs, tumor-cell-derived exosomes; DEX, DC-derived exosomes; DEX-AFP, exosomes derived from AFP-expressing DCs; CAR-T, chimeric antigen receptor T; HSP, heat shock protein; NK, natural killer cell; M1, M1-polarized macrophage; MSC, mesenchymal stem cell; NKT, natural killer T cell.

**Table 1 cancers-14-04036-t001:** Potential role of exosomes in improving the efficacy of HCC immunotherapies.

Type of Treatment	Exosomes	Role	References
ICIs	PD-L1	Administrate miR-220a/220b/429 mimics to interference with HMGB1 and RICTOR, thereby inhibiting the production of PD-L1-contaning exosomes	[88]
	PD-L1	Use hemofiltration to remove the PD-L1-contaning exosomes from the bloodstream	[91]
	HCC-derived exosomes	Administrate exosomes derived from HCC cells treated with 0.1 Mm melatonin, thereby downregulating the expression of PD-L1 on macrophages	[92]
	miR-15a-5p	Inhibit PD-1 expression on CD8+ T cells	[31]
	Exosome-encapsulated small interfering RNA	Target β-catenin, thereby blocking Wnt/β-catenin signaling, which can contribute to immune evasion	[93]
	DC-TEX	Combine DC-TEX and PD-1 inhibitor to enhance the efficacy of sorafenib	[96]
	CTLA-4, PD-1	Use tonifying traditional Chinese medicine to treat spleen deficiency, thereby reducing the exosomal CTLA-4 and PD-1	[85]
DC vaccine	TEXs	Carry HCC antigens and trigger a strong DC- mediated immune response	[37]
	TEX-N1ND	Strengthen DC immunogenicity and suppress large established tumors	[99]
	DEX	Promote the proliferation of naïve T cells and differentiate to cytotoxic T lymphocytes	[39]
	DEX-AFP	Increase the number of CD8+ T cells and reduce the number of CD25+ Foxp3+ regulatory T cells	[38]
Virus vaccine	EVM/VSV-G Ad5-P	Enhance the efficacy of type V adenovirus	[101]
CAR-T	RN7SL1	Co-deploy peptide antigen and enhance the efficacy of CAR-T	[104]
	CAR-containing exosomes	Hold potential anti-tumor effects	[105]
	CAR-T cell-derived exosomes	Can be positive modulated by T lymphocyte activation enhancers	[106]
NK cell	HSP-bearing exosomes	Elicit efficient NK cells	[109]
	Exosomes derived from hepatoma G2 cells	Reinforce the cytotoxicity of NK cells	[110]
M1 macrophage	Exosomes	Synergize with PIONs@E6 and heighten the M1 macrophages polarization	[111]
	Exosomes derived from M1 macrophages (IL4R-Exo)	Heighten the M1 macrophages polarization	[112]
	Engineered exosomes (exoASO-STAT6)	Silence STAT6 expression in tumor-associated macrophages and reprogram them to M1 phenotype	[113]
Others	miR-125b-loaded EVs	Specifically reduce HCC cell proliferation by regulating the p53 signaling pathway	[115]
	EVs from red blood cells	Accumulate in liver and diminish systemic toxicity of delivered drugs	[116]
Mesenchymal stem cells	Homologous TEXs	Enhance the migratory capacity of bone MSCs, which have great antitumor activities	[117]
	BMSC-derived exosomal miR-338-3p	Down-regulate EST1 and thereby inhibit HCC	[118]
	Adipose-derived MSCs exosomes	Promote the anti-tumor response of NKT cells	[119]
	Human umbilical cord MSCs exosomal miR-451a	Restrict the epithelial-mesenchymal transition of HCC cells	[120]

Developing drugs for exosomes could enhance the efficacy of ICIs, tumor vaccines, adoptive cell therapy, and others. ICIs, immune checkpoint inhibitors; DC, dendritic cell; CAR-T, chimeric antigen receptor T; NK, nature killer; PD-L1, programmed cell death ligand 1; CTLA-4, cytotoxic-T-lymphocyte-associated protein 4; PD-1, programmed cell death protein 1; DC-TEX, TEX-pulsed DCs; DEX, DC-derived exosomes; DEX-AFP, exosomes derived from AFP-expressing DCs; EVs, extracellular vesicles; TEX, tumor-cell-derived exosomes; MSCs, mesenchymal stem cells; HCC, hepatocellular carcinoma.

**Table 2 cancers-14-04036-t002:** Exosomes in predicting the efficacy of immunotherapy.

Detection Method	
Traditional detection technology	Use ultracentrifugation to separate the exosomes; use RT-PCR to evaluate the exosomal miRNA; use mass spectrometry or ELISA to assess exosomal protein
Novel detection technology	Use acoustic tweezer techniques in combination with microfluidics or commercial kits to isolate the exosomes
**Potential samples for** **detection**	
Source of body fluids	Blood (plasma or serum), ascites, and bile (may be the most appropriate), etc.
**Possible predictive** **biomarkers**	
Exosomal PD-L1	Has been demonstrated in other tumors and exosomal PD-L1 may better than other forms of extracellular PD-L1
Exosomal genes MYL6B and THOC2	Influence the expression of immune checkpoint genes
Exosomal miR-143-3p	Upregulate the expression of MARCKS in TAMs
Exosomal HMGB1	The prediction of HGMB1 for HCC prognosis has been clinically confirmed

PD-L1, programmed cell death ligand 1; ELISA, enzyme-linked immunosorbent assay; TAMs, tumor-associated macrophages; HCC, hepatocellular carcinoma.

## Data Availability

All data and material generated or analyzed during this study are included in this published article.

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
