# Peer review of "Role of Exosomes in Immunotherapy of Hepatocellular Carcinoma"

_cancers, 2022, doi:10.3390/cancers14164036_

Round 1
Reviewer 1 Report
The review by Tian et al on exosomes is a well written and informative compendium on what is known on exosomes in HCC both from the tumor perspective and as part of immunotherapies. The content is complemented with will designed figures that bring a lot of information in a well presented way. It is very complete and has only a few typos, minor mistakes that will need to be corrected.
*I would consider revising the third sentence in the abstract (which starts in line 11). You cannot give immunotherapy therefore affects immunotherapy?
* I don't understand what they mean with the sentence in line 21. Are they reviewing the targeting of exosomes or using exosomes as therapies? Please clarify this sentence.
* Line 40, CTLA-4 not CTAL-4.
*I would remove the first sentence in section 2.1 line 76. It is either out of place, irrelevant or both.
*Line 119, ICI have not been defined at that point. Please define here.
*Line 133 Change to CTLA-4.
*Line 148 define TGN.
*Figure 3 says CTAL instead of CTLA.
Author Response
Dear editor and review:
Thank you so much for your response to our manuscript (Manuscript ID: cancers-1784171), entitled “Role of Exosomes in Immunotherapy of Hepatocellular Carcinoma”. We also appreciate the constructive and enthusiastic as well as fare comments from the reviewers. The reviewer’s comments have been most useful to us in the preparation of a more concise manuscript. Accordingly, we have revised the enclosed manuscript based on the reviewer’s suggestions. All major revisions were labeled in the revised manuscript.
Specific responses to reviewer’s comments are described below.
Comment 1: I would consider revising the third sentence in the abstract (which starts in line 11). You cannot give immunotherapy therefore affects immunotherapy?
Response: We appreciate the reviewer for the constructive comments. We have revised that sentence in the abstract to make it clearer and more accurate.
Comment 2: I don't understand what they mean with the sentence in line 21. Are they reviewing the targeting of exosomes or using exosomes as therapies? Please clarify this sentence.
Response: We are very sorry for the inconvenience caused to the reviewer. Some studies applied exosomes as therapeutic targets and others applied exosomes to improve the efficacy of immunotherapy. We have revised this sentence in the abstract so that readers can gain a better understanding of our articles through the abstract
Comment 3: Line 40, CTLA-4 not CTAL-4.
Response: We are sorry for the mistake. We have corrected this word in the revised manuscript.
Comment 4: I would remove the first sentence in section 2.1 line 76. It is either out of place, irrelevant or both.
Response: As the reviewer suggested, we have deleted the first sentence in section 2.1.
Comment 5: Line 119, ICI have not been defined at that point. Please define here.
Response: We have corrected the mistake in the revised manuscript and defined ICIs when it was first used.
Comment 6: Line 133 Change to CTLA-4.
Response: We have corrected the mistake in the revised manuscript. We have carefully checked the manuscript and made English corrections.
Comment 7: Line 148 define TGN.
Response: As the reviewer suggested, we have changed TGN into trans-Golgi network (TGN).
Comment 8: Figure 3 says CTAL instead of CTLA.
Response: We are very sorry for the mistake. We have revised the Figure 3 according to suggestion.
We have now responded to all the comments raised by the reviewer. We hope reviewers find our revised manuscript acceptable and the paper meets the editorial requirements. We look forward to your expert advice as well as the constructive comments of the reviewers.
Sincerely yours
Tao Li
Department of general surgery, Qilu Hospital, Shandong University
107 West Wen Hua Road
Jinan, 250012
China.
People’s Republic of China
Tel & Fax: +86-531-82166651
Email: litao7706@163.com
Reviewer 2 Report
The paper entitled “Role of Exosomes in Immunotherapy of Hepatocellular Carcinoma” by Bao-Wen Tian et al., is a review in which the authors summarize and discuss the potential of exosomes as biomarkers for predicting the efficacy of hepatocellular carcinoma immunotherapy, summarizing their role on tumor microenvironment of hepatocellular carcinoma.
The topic is very interesting and the organization of the paragraphs too.
However, the manuscript needs to be edited for English language, there are several formal mistakes and excessive use of colloquial language.
It could also be useful to describe the method used to select the papers discussed in the review, for example how many papers, the range of years considered, the keywords used to make the research, etc.
Some titles should be revised in general, for example titles of paragraphs 3 and 3.1 are too similar.
Author Response
Dear editor and review:
Thank you so much for your response to our manuscript (Manuscript ID: cancers-1784171), entitled “Role of Exosomes in Immunotherapy of Hepatocellular Carcinoma”. We also appreciate the constructive and enthusiastic as well as fare comments from the reviewers. The reviewer’s comments have been most useful to us in the preparation of a more concise manuscript. Accordingly, we have revised the enclosed manuscript based on the reviewer’s suggestions. All major revisions were labeled in the revised manuscript.
Specific responses to reviewer’s comments are described below.
Comment 1: The manuscript needs to be edited for English language, there are several formal mistakes and excessive use of colloquial language.
Response: Thank for the reviewer suggestion, we have carefully checked and revised our manuscript to make it more concise and smoother. In addition, we also used the editor recommended language editing services to improve the English language of the revised manuscript.
Comment 2: It could also be useful to describe the method used to select the papers discussed in the review, for example how many papers, the range of years considered, the keywords used to make the research, etc.
Response: We are very sorry that we do not keep a detailed record of our strategy for each search and the number of documents retrieved. At the same time, in some sections, we only keep the relatively high qualitied literature with strong relevance to our topic. In each of our retrieval processes, we do not limit the range of years, article types, and other conditions. Our main search terms include but are not limited to ‘hepatocellular carcinoma’, ‘exosomes’, ‘immunotherapy’ and ‘immune checkpoint inhibitors’, etc. We have added some explanation in the revised manuscript.
Comment 3: Some titles should be revised in general, for example titles of paragraphs 3 and 3.1 are too similar.
Response: We appreciated the reviewer for the constructive comments. We have revised all inappropriate titles in our revised manuscript.
We have now responded to all the comments raised by the reviewer. We hope reviewers find our revised manuscript acceptable and the paper meets the editorial requirements. We look forward to your expert advice as well as the constructive comments of the reviewers.
Sincerely yours
Tao Li
Department of general surgery, Qilu Hospital, Shandong University
107 West Wen Hua Road
Jinan, 250012
China.
People’s Republic of China
Tel & Fax: +86-531-82166651
Email: litao7706@163.com
Reviewer 3 Report
The authors set out to review the role of exosomes in hepatocellular carcinoma (HCC), their role in the immunotherapy of HCC and the possibility to predict the efficacy of immunotherapy of HCC based on the presence of specific exosomes.
The topic is interesting. However perhaps the authors should have focussed on one of these aspects to provide a more comprehensive and clear review.
The manuscript is not well written, difficult to follow and would require extensive editing. The text is not so much a ‘story’ but rather an enumeration of loose sentences referring to a diversity of studies with no, or limited, links between them.
In my view the authors should seriously consider to perform a more thorough study of the literature and write a new, more focussed and detailed review.
Yet despite this, herewith some, but surely not a comprehensive list of comments and suggestions.
1. The abstract is very unclear in describing the goal of the review. Needs to be rewritten entirely.
2. The manuscript largely focusses on exosomal PD-L1. Currently, many efforts have been devoted to improve the efficacy of anti-PD-L1 therapy, mainly via (anti PD-LI + X). However, with small progress so far. Do the authors envision that the exosomal PD-L1 ‘discovery’ contributes to our knowledge on basic adaptive immune resistance mechanisms which will guide more efficacious cancer immunotherapies in the future, instead of another exhaustive combination of anti PD-LI + X?
3. In the section - 2.1 Influence of Exosomes on Cells in Tumor Microenvironment of HCC. It would be interesting if authors could show details about which tumor cell-derived exosomes could differentiate the macrophages towards M1/M2, and the influences of M1/M2 macrophages derived exosomes on tumors. Furthermore, in terms of the influences of exosomes on immune cells, perhaps B cells, Tregs and MDSCs should be discussed, given their vital roles in the TME.
4. In the section – 2.2. Possible Mechanism of Exosomes on Immunotherapy Resistance of HCC. The authors addressed possible mechanism of exosomes in terms of immune checkpoint blockade therapy resistance. I wonder if exosomes can also mediate therapy resistance in other immunotherapies such as cancer vaccination, adoptive cell therapies and antibody-drug conjugate et al. For instance, would the signal transmission between therapy-resistant cells to therapy-sensitive cells impair the above immunotherapies?
5. Line 198 – Are there any disadvantages when removing the circulating exosomes from the bloodstream using hemofiltration? Could this method allow to selectively remove immunosuppressive exosomes, without removing other exosomes responsible for the normal intercellular communication?
6. The authors mentioned that exosomal PD-L1 could be used to predict the prognosis or response to immunotherapy in various cancer types. Please discuss/address if exosomal PD-L1 is better than the current clinical cancer markers and the PD-L1 expression on circulating tumor cells in terms of prognosis, recurrence, and the efficacy of immunotherapy predictions?
Minor comments
7. In Figure 1, HCC is commonly used as the abbreviation of ‘hepatocellular carcinoma’, not for ‘hepatocellular carcinoma cell’. The exact meaning of “regulation” and “contribution” in the figure should be better illustrated. Changing Figure 1 into a list would possibly make it more clear to readers.
8. Authors mention the exosomes containing HMGB1 exert an essential role in mediating HCC resistance to immunotherapies. It would need to be noted that HMGB1 appears to play paradoxical roles during the development and therapy of cancer, apart from contributing to tumorigenesis, HMGB1 activates the immune response and plays a protective role in the suppression of tumors and immunotherapy.
9. Line 40: correct CTLA-4
10. Line 88 – ‘HCC-derived exosomes” should be ‘tumor cell-derived exosomes’
11. Line 245- ‘CD35+Foxp3+ regulatory T cells’ should be corrected for ‘‘CD25+Foxp3+ regulatory T cells’.
12. Line 246 – ‘DEX-ATP’ should be corrected for ‘DEX-AFP’
13. Line 345: ‘not entire patients benefit from…” ??
14. And many more typo’s and incorrect or non-scientific expressions.
Author Response
Dear editor and review:
Thank you so much for your response to our manuscript (Manuscript ID: cancers-1784171), entitled “Role of Exosomes in Immunotherapy of Hepatocellular Carcinoma”. We also appreciate the constructive and enthusiastic as well as fare comments from the reviewers. The reviewer’s comments have been most useful to us in the preparation of a more concise manuscript. Accordingly, we have revised the enclosed manuscript based on the reviewer’s suggestions. All major revisions were labeled in the revised manuscript.
Specific responses to reviewer’s comments are described below.
Comment 1: The abstract is very unclear in describing the goal of the review. Needs to be rewritten entirely.
Response: Many thanks to the reviewer for the constructive comments. We have rewritten our abstract. In addition, we also used the editor recommended language editing services to improve the English language of the revised manuscript.
Comment 2: The manuscript largely focusses on exosomal PD-L1. Currently, many efforts have been devoted to improve the efficacy of anti-PD-L1 therapy, mainly via (anti PD-LI + X). However, with small progress so far. Do the authors envision that the exosomal PD-L1 ‘discovery’ contributes to our knowledge on basic adaptive immune resistance mechanisms which will guide more efficacious cancer immunotherapies in the future, instead of another exhaustive combination of anti PD-LI + X?
Response: Thanks to the reviewer for the constructive comments. The first defined and therapeutically validated adaptive immune resistance (AIR) mechanism is the selective induction of PD-L1 by interferon-γ in the tumor. As the reviewer pointed out, antibody blockade of PD-L1 can activate an anti-tumor immune response leading to durable remissions in a subset of cancer patients. However, many clinical trials combining anti-PD therapy with other antitumor drugs have failed to identify a synergistic or additive effect.
It is generally thought that PD-L1 functions within the tumor bed, where cell-surface PD-L1 is directly interacting with PD-1 on the surface of TILs. However, recent studies discover that tumor cells can secrete a vast majority of their PD-L1 on exosomes rather than present PD-L1 on their cell surface. Exosomal PD-L1 enables cancer cells to evade anti-tumor immunity and appears to be resistant to anti-PD-L1 antibody blockade. Removal of exosomal PD-L1 inhibits tumor growth, even in models resistant to anti-PD-L1 antibodies. However, the degree to which exosomal versus surface PD-L1 in driving immunosuppression will vary between patients and cancer types, and will be critical in deciding who are more likely to respond to therapy. Therefore, measuring both cell-surface and exosome presentation of PD-L1 should be considered in any therapeutic strategy, and it will be interesting to determine whether localized anti-exosomal therapy combined with systemic anti-PD-L1 blockade could synergize to induce a systemic immune response against tumor.
As the reviewer said, we believe that exosomal PD-L1 ‘discovery’ contributes to our knowledge on basic adaptive immune resistance mechanisms which will guide more efficacious cancer immunotherapies in the future, instead of another exhaustive combination of anti PD-LI + X. We have added the above discussion in the revised manuscript.
Comment 3: In the section - 2.1 Influence of Exosomes on Cells in Tumor Microenvironment of HCC. It would be interesting if authors could show details about which tumor cell-derived exosomes could differentiate the macrophages towards M1/M2, and the influences of M1/M2 macrophages derived exosomes on tumors. Furthermore, in terms of the influences of exosomes on immune cells, perhaps B cells, Tregs and MDSCs should be discussed, given their vital roles in the TME.
Response: Many thanks to the reviewer for the constructive suggestions. We have reconsidered all relevant literatures, supplemented the required literatures in our manuscript, revised the pictures, and added Supplementary Table 1 to facilitate readers' better understanding of the relevant content. Details about the relationship between exosomes and M1/M2 macrophages have been described in Supplementary Table 1. In the study of HCC exosomes, there are relatively few studies has been taken on B cells, Tregs and MDSCs, and we have discussed the related articles in the main text.
Comment 4: In the section – 2.2. Possible Mechanism of Exosomes on Immunotherapy Resistance of HCC. The authors addressed possible mechanism of exosomes in terms of immune checkpoint blockade therapy resistance. I wonder if exosomes can also mediate therapy resistance in other immunotherapies such as cancer vaccination, adoptive cell therapies and antibody-drug conjugate et al. For instance, would the signal transmission between therapy-resistant cells to therapy-sensitive cells impair the above immunotherapies?
Response: The current research on exosome-mediated HCC immunotherapy resistance focuses on immune checkpoint inhibitors, and there is no direct related research on the resistance mechanism of other immunotherapy, such as cancer vaccination and adoptive cell therapies et al. Considering the impact of exosomes on the immune microenvironment of HCC, exosomes are likely to be involved in other immunotherapy resistance mechanisms, which are warranted to clarified by further researches. We have added some discussion in the revised manuscript.
Comment 5: Line 198 – Are there any disadvantages when removing the circulating exosomes from the bloodstream using hemofiltration? Could this method allow to selectively remove immunosuppressive exosomes, without removing other exosomes responsible for the normal intercellular communication?
Response: Circulating exosomal PD-L1 has the role in mediating systemic immunosuppression, and thus there may have the risk of serious immune-related adverse events after eliminated them, although some preclinical studies have not found obvious irAEs. Hemofiltration indeed can allow to selectively remove immunosuppressive exosomes, without removing other exosomes responsible for the normal intercellular communication. In this approach, we can customize the component of matrices to specifically capture and remove soluble proteins from plasma. For instance, PD-L1 positive exosomes can be eliminated by blood purification devices equipped with a matrix containing anti-PD-L1 antibody. We have added the following discussion in the revised manuscript:
Although hemofiltration can selectively remove immunosuppressive exosomes without removing other exosomes responsible for the normal intercellular communication, and some preclinical studies have not found obvious immune-related adverse events, however, it should be noted that, there may be the risk of serious immune-related adverse events after eliminating circulating exosomal PD-L1, since it plays a role in mediating systemic immunosuppression.
Comment 6: The authors mentioned that exosomal PD-L1 could be used to predict the prognosis or response to immunotherapy in various cancer types. Please discuss/address if exosomal PD-L1 is better than the current clinical cancer markers and the PD-L1 expression on circulating tumor cells in terms of prognosis, recurrence, and the efficacy of immunotherapy predictions?
Response: Until know, no studies have directly compared the predictive efficiency of exosomal PD-L1 with other clinical tumor markers, and some studies have found that exosomal PD-L1 cooperating with other tumor markers or immune modulators (such as CD28), may be a better predictive biomarker than exosomal PD-L1 alone. Though PD-L1 expression on circulating tumor cells (CTCs) may be predictive of response to immunotherapy. Unfortunately, the detection of CTCs remains a technical challenge especially as CTCs are usually very rare and different methods might enrich different CTC populations, which will in turn affect the PD-L1 assessment. In contrast, exosomes are stable and abundant in plasma, and can be quantified by ELISA. Therefore, exosomal PD-L1 may be the most promising potential biomarker in this area based on current relevant pan-cancer studies. However, since the predictive value of exosomal PD-L1 and PD-L1 on CTCs may be different in different tumors, relevant studies are needed to explore their application potential in HCC. We have added the above-mentioned discussion in the revised manuscript.
Minor comments
Comment 7: In Figure 1, HCC is commonly used as the abbreviation of ‘hepatocellular carcinoma’, not for ‘hepatocellular carcinoma cell’. The exact meaning of “regulation” and “contribution” in the figure should be better illustrated. Changing Figure 1 into a list would possibly make it more clear to readers.
Response: We appreciate the reviewer for the constructive comments. As the reviewer suggested, we have changed ‘hepatocellular carcinoma cell’ into ‘hepatocellular carcinoma’ in the Figure 1. We also modified the same inappropriate representations in the rest of the Figures. What’s more, we have added Supplementary Table 1 which contain all known exosomes-mediated cell-to-cell communication pathways in HCC tumor microenvironment, including the exact meaning of “regulation” and “contribution”, to facilitate readers to gain a better understanding of our mechanism diagram and relevant contents.
Comment 8: Authors mention the exosomes containing HMGB1 exert an essential role in mediating HCC resistance to immunotherapies. It would need to be noted that HMGB1 appears to play paradoxical roles during the development and therapy of cancer, apart from contributing to tumorigenesis, HMGB1 activates the immune response and plays a protective role in the suppression of tumors and immunotherapy.
Response: We appreciate the reviewer for the constructive comments. As the reviewer state, the roles of HMGB1 in tumors are complex and contradictory. We have added the following discussion in the revised manuscript:
However, it should be noted that HMGB1 appears to play paradoxical roles during the development and therapy of cancer, and its value as potential target for therapies is worthy of further study.
Comment 9: Line 40: correct CTLA-4
Response: We are very sorry for the mistake. We have corrected the mistake in the revised manuscript.
Comment 10: Line 88 – ‘HCC-derived exosomes” should be ‘tumor cell-derived exosomes’
Response: As the reviewer suggested, we have changed ‘HCC-derived exosomes’ into ‘tumor cell-derived exosomes’ in our revised manuscript.
Comment 11: Line 245- ‘CD35+Foxp3+ regulatory T cells’ should be corrected for ‘‘CD25+Foxp3+ regulatory T cells’.
Response: We are sorry for the mistake. We have corrected the mistake in the revised manuscript.
Comment 12: Line 246 – ‘DEX-ATP’ should be corrected for ‘DEX-AFP’
Response: We are sorry for the mistake. We have corrected the mistake in the revised manuscript.
Comment 13: Line 345: ‘not entire patients benefit from…” ??
Response: We are sorry for the mistake. We have revised that sentence in our revised manuscript.
Comment 14: And many more typo’s and incorrect or non-scientific expressions.
Response: We have corrected the mistake in the revised manuscript. We have carefully checked the manuscript and made English corrections. In addition, we also used the editor recommended language editing services to improve the English language of the revised manuscript.
We have now responded to all the comments raised by the reviewers. We hope reviewers find our revised manuscript acceptable and the paper meets the editorial requirements. We look forward to your expert advice as well as the constructive comments of the reviewers.
Sincerely yours
Tao Li
Department of general surgery, Qilu Hospital, Shandong University
107 West Wen Hua Road
Jinan, 250012
People’s Republic of China
Tel & Fax: +86-531-82166651
Email: litao7706@163.com
Reviewer 4 Report
Congratulations to the authors for the interesting and clear review. This review is well described and summarizes the knowledge on the future immunotherapeutic treatment of the HCC.
There are very little minor issues that should be considered:
Please replace the size 200 nm of exosomes with 150 nm in the abstract and at line 50 of the text and here add also a reference (i.e. van Niel et al., 2018);
Line 92, please specify what TS/A means;
Line 227, please move the tile 3.2 “Role of……….” after Figure 4.
Author Response
Dear editor and review:
Thank you so much for your response to our manuscript (Manuscript ID: cancers-1784171), entitled “Role of Exosomes in Immunotherapy of Hepatocellular Carcinoma”. We also appreciate the constructive and enthusiastic as well as fare comments from the reviewers. The reviewer’s comments have been most useful to us in the preparation of a more concise manuscript. Accordingly, we have revised the enclosed manuscript based on the reviewer’s suggestions. All major revisions were labeled in the revised manuscript.
Specific responses to reviewer’s comments are described below.
Comment 1: Please replace the size 200 nm of exosomes with 150 nm in the abstract and at line 50 of the text and here add also a reference (i.e. van Niel et al., 2018);
Response: We appreciate the reviewer for the constructive comments. As the reviewer suggested, we have changed ‘200nm’ into ‘150nm’ and add the relevant reference (van Niel et al., 2018) in our revised manuscript.
Comment 2: Line 92, please specify what TS/A means;
Response: TS/A is a name of the cell line of murine mammary adenocarcinoma. Because our topic focused on HCC, we deleted this sentence from the revised manuscript.
Comment 3: Line 227, please move the tile 3.2 “Role of……….” after Figure 4.
Response: As the reviewer suggested, we have moved the title of Section 3.2 after the Figure4 in our revised manuscript.
We have now responded to all the comments raised by the reviewers. We hope reviewers find our revised manuscript acceptable and the paper meets the editorial requirements. We look forward to your expert advice as well as the constructive comments of the reviewers.
Sincerely yours
Tao Li
Department of general surgery, Qilu Hospital, Shandong University
107 West Wen Hua Road
Jinan, 250012
People’s Republic of China
Tel & Fax: +86-531-82166651
Email: litao7706@163.com